# BC^2^NetRF: Breast Cancer Classification from Mammogram Images Using Enhanced Deep Learning Features and Equilibrium-Jaya Controlled Regula Falsi-Based Features Selection

**DOI:** 10.3390/diagnostics13071238

**Published:** 2023-03-25

**Authors:** Kiran Jabeen, Muhammad Attique Khan, Jamel Balili, Majed Alhaisoni, Nouf Abdullah Almujally, Huda Alrashidi, Usman Tariq, Jae-Hyuk Cha

**Affiliations:** 1Department of Computer Science, HITEC University, Taxila 47080, Pakistan; kiran.jabeen@hitecuni.edu.pk (K.J.); attique.khan@hitecuni.edu.pk (M.A.K.); 2College of Computer Science, King Khalid University, Abha 61413, Saudi Arabia; jabaili@kku.edu.sa; 3Higher Institute of Applied Science and Technology of Sousse (ISSATS), Cité Taffala (Ibn Khaldoun) 4003 Sousse, University of Souse, Sousse 4000, Tunisia; 4College of Computer Science and Engineering, University of Ha’il, Ha’il 81451, Saudi Arabia; m.alhaisoni@uoh.edu.sa; 5Department of Information Systems, College of Computer and Information Sciences, Princess Nourah bint Abdulrahman University, P.O. Box 84428, Riyadh 11671, Saudi Arabia; 6Faculty of Information Technology and Computing, Arab Open University, Ardiya 92400, Kuwait; halresheedi@aou.edu.kw; 7Department of Management, CoBA, Prince Sattam bin Abdulaziz University, Al-Kharj 11942, Saudi Arabia; u.tariq@psau.edu.sa; 8Department of Computer Science, Hanyang University, Seoul 04763, Republic of Korea; chajh@hanyang.ac.kr

**Keywords:** breast cancer, mammogram images, contrast enhancement, augmentation, deep learning, feature optimization, feature fusion, neural networks

## Abstract

One of the most frequent cancers in women is breast cancer, and in the year 2022, approximately 287,850 new cases have been diagnosed. From them, 43,250 women died from this cancer. An early diagnosis of this cancer can help to overcome the mortality rate. However, the manual diagnosis of this cancer using mammogram images is not an easy process and always requires an expert person. Several AI-based techniques have been suggested in the literature. However, still, they are facing several challenges, such as similarities between cancer and non-cancer regions, irrelevant feature extraction, and weak training models. In this work, we proposed a new automated computerized framework for breast cancer classification. The proposed framework improves the contrast using a novel enhancement technique called haze-reduced local-global. The enhanced images are later employed for the dataset augmentation. This step aimed at increasing the diversity of the dataset and improving the training capability of the selected deep learning model. After that, a pre-trained model named EfficientNet-b0 was employed and fine-tuned to add a few new layers. The fine-tuned model was trained separately on original and enhanced images using deep transfer learning concepts with static hyperparameters’ initialization. Deep features were extracted from the average pooling layer in the next step and fused using a new serial-based approach. The fused features were later optimized using a feature selection algorithm known as Equilibrium-Jaya controlled Regula Falsi. The Regula Falsi was employed as a termination function in this algorithm. The selected features were finally classified using several machine learning classifiers. The experimental process was conducted on two publicly available datasets—CBIS-DDSM and INbreast. For these datasets, the achieved average accuracy is 95.4% and 99.7%. A comparison with state-of-the-art (SOTA) technology shows that the obtained proposed framework improved the accuracy. Moreover, the confidence interval-based analysis shows consistent results of the proposed framework.

## 1. Introduction

Cancer affects people all over the world. Breast cancer is the most prevalent type among women [1]. However, according to a Breast Cancer Care (BCC) survey, 42% of NHS trusts claim they do not have enough staff to assign people, citing minimal specialist nursing experience in breast cancer. It is the main factor that contributes to the low global survival rate of breast cancer [2]. A shortage of breast cancer specialists among medical professionals will delay disease detection, encourage noncompliance with the best screening and treatment methods, and lead to unequal access to the best care [3]. Breast cancer detection was established to effectively detect abnormalities and categorize breast cancer. This is done to help in breast cancer diagnosis [4]. Early detection is crucial to lowering the death rate; however, due to the modest sizes of possible nodules concerning the overall breast, early diagnosis of breast cancer by screening mammography is difficult [5]. Compared to other cancer types, breast cancer has the highest probability of being treated (around 90%). Cancer does not produce early pain, so it is not noticed until serious health issues arise [6]. Age is a factor in both breast cancer mortality and incidence rates. The average age of breast cancer diagnosis between 2010 and 2014 is 62.

Pakistan has Asia’s highest breast cancer incidence rate—annual reports of cases totaling about 90,000 results include a fatality rate of 40,000 [7]. The patient survival rate is the percentage of patients expected to live after a diagnosis for a pre-determined amount of time with the expectation of leading a normal life. The stage at which cancer is discovered affects the survival rate [8]. Due to its ability and low cost to satisfy medical requirements, mammography has appeared as the most reliable tool for detecting breast cancer.

Mammography analysis is the doctor’s primary method to determine a diagnosis, but this method is vulnerable to bias and doctor tiredness [1]. Unfortunately, mammography has a low detection rate. It can produce false-negative results in the range of 5% to 30%, based on the type of lesion, the density of the breasts and the patient’s age [9]. Therefore, low-dose radiography is used in mammography, which allows us to see the inside structure of the breast. Various signal processing methods are used to detect breast cancer, including ultrasound imaging [10], microwave imaging [11], and curvelet transform [8,12].

Computer-aided detection (CAD) technologies for breast cancer are crucial therapeutically in order to lessen radiologists’ labor and increase their detection precision [5]. Pattern recognition is the foundation of the conventional method for classifying medical infections, such as breast masses, skin lesions, and brain tumors. For breast cancer, the mammogram features are manually extracted, and the extracted features are then entered into a machine learning classifier for categorization [1]. However, gaining an accurate classification is still challenging due to several image issues and variations in the tumor regions. Therefore, AI has been significantly involved in detecting and classifying medical infections in the last decade, especially for breast cancer [13].

Thanks to the CAD system, this is based on several intermediate stages, such as the preprocessing of original images, feature learning and extraction, feature selection and reduction, and classification [14]. In the preprocessing step, the researcher attempts to produce good-quality images and overcome the noise if it exists. The objective of preprocessing is to make the tumor region more visible, which can later help in accurately detecting a region of interest (ROI) [15]. Several conventional approaches have been described in the literature for ROI detection, such as fuzzy, clustering, and saliency-based techniques, to name but a few. The next important step is the feature extraction, in which each image’s key properties are computed. Several conventional feature extraction techniques, such as the shape, texture, and point features, are introduced in the literature. Some researchers also focused on feature reduction and selection for improved accuracy and less computational time [16]. The last important step in AI is the machine learning techniques based on classifying cancer regions into relevant categories, such as cancerous or non-cancerous [17].

Recently, a convolutional neural network (CNN) has shown a remarkable performance in medical imaging for detecting and classifying cancers. The effectiveness of the deep learning models usually depends on the size of the training datasets. The conventional techniques did not perform well for the complex nature datasets; however, on the other side, the deep learning-based techniques showed impressive performance. Truly, deep learning uses the idea of CNN to classify breast cancer. A CNN model contains layers (hidden layers), such as convolutional, pooling, activation, and fully connected layers. The last layer of a CNN model is Softmax, which works as a classifier. Deep learning empowers automated AI techniques in medical imaging. Several deep learning-based architectures have been introduced in the literature for diagnosing and classifying medical infections [13]. The researchers introduced many deep learning approaches for breast cancer classification and diagnosis; however, they still face challenges, such as imbalanced datasets, noisy imaging data, and the downsampling of important features [18]. They focused on the learning task of deep models based on transfer learning. Transfer learning is a method of reusing a pre-trained model for another purpose or task [19]. Several hyperparameters have been employed for the training process, such as the learning rate and mini-batch size, etc.; however, for breast cancer, it is difficult to initialize the manual values of each parameter. Obayya et al. [20] presented an optimized hyperparameter-based deep-learning framework for classifying breast cancer. After the training, the researcher extracted the deep features from the fully connected layer; however, based on the analysis, it was noticed that several features were redundant and affected the classification accuracy of breast cancer [21]. Recently, Atban et al. [22] presented an optimized deep learning approach for improved breast cancer classification. Peirera et al. [23] presented a dialectical feature selection technique for enhanced breast cancer classification; however, these techniques face the problem of termination where the optimal values have been retrieved.

### 1.1. Major Challenges and Contributions

Several challenges exist in the medical image processing domain, especially for breast cancer classification when using deep learning. The first challenge is fewer amounts of available mammography image datasets because a deep learning model requires a larger dataset for the training and a better understanding of the classification task [24]. The second challenge is feature engineering. This step also extracts several redundant features that cause a false-negative rate and high computational time [25]. In this paper, we proposed a new framework based on original and enhanced mammogram images’ deep-learning optimal features aggregation. Our main contributions to this work are listed as follows:We proposed a contrast enhancement approach based on the haze reduction concept called haze-reduced local-global (HRLG).We performed data augmentation and trained the fine-tuned EfficientNet-b0 deep learning model. For the training, hit- and trial-based values were selected for the hyperparameters of the network. This model was trained using original and enhanced images and extracted deep features from the average pool layer instead of a fully connected layer. The extracted deep features were later fused using a serial nature approach.A feature selection technique is proposed and called Equilibrium-Jaya controlled Regula Falsi. The best features are selected and fused using a new, short serial-based technique.

### 1.2. Justification of the Contributions

The major objective of the proposed contrast enhancement strategy is to increase the local information of the image, which is later helpful in the better learning of a deep model and improved classification (the results-based justification is provided under Section 4). The reason behind the selection of EfficientNet-b0 is based on the lesser number of parameters and a better top-five accuracy. In addition, the hit- and trial-based hyperparameters selection shows better model learning that increases the training accuracy. In addition, the proposed feature selection technique reduces the problem of overfitting and lessens the proposed framework’s computational cost (the results are given in Section 4).

The remainder of the article is arranged in the following order. Section 2 defines the related work of this manuscript. The proposed methodology, discussed in Section 3, includes the proposed contrast enhancement, augmentation, feature extraction, optimization, fusion, and classification. The outcomes, in the form of numerical values, are presented in Section 4. Finally, Section 5 concludes the manuscript.

## 2. Related Work

The most common type of cancer worldwide is breast cancer, with about 1.7 million women affected by cancer being reported in 2012. Age, family history, and medical history are just a few risk factors for breast cancer [4]. Most patients who die from cancer are women, and 2.1 million of them are identified with breast cancer each year. According to a recent survey, 627,000 women are estimated to have died from cancer in 2018, making up 15% of all cancer fatalities among women [5]. Breast cancer detection and its classification via computer visualization are commonly performed through a deep learning-based model. However, due to the complexity of early breast cancer and the dimming of mammography images, it is challenging for clinicians to diagnose cancer from these images. As a result, it is crucial to improve a doctor’s detection effectiveness using the CAD system of deep learning approaches [26].

Tan et al. [4] suggested a CNN-based framework for classifying mammography images into normal, benign, and malignant to classify breast cancer. First, to visualize the mammogram images, preprocessing was performed. Then, the preprocessed images were trained on the deep learning model that extracted the features. The extracted features of the last layer were finally classified using a CNN classifier called Softmax. The selected model increased the classification accuracy of mammography images using the introduced framework. The results clearly show that the suggested framework was more accurate than other existing methods, having an accuracy of 0.8585 and 0.8271, respectively. Falconi et al. [27] presented the preliminary findings for the classification of breast abnormalities as malignancies using transfer learning. They used several deep learning models and found two top models, ResNet50 and MobileNet. Both models produced the best outcomes, with 78.4% and 74.3% accuracy, respectively. Additionally, they also applied several preprocessing techniques to improve the classification’s precision. Finally, Samee et al. [28] presented a novel hybrid processing method, which is based on both principal component analysis (PCA) and logistic regression (LR).

The CAD system was evaluated using the INbreast and mini-MIAS benchmark datasets. The presented CAD system achieved the best accuracies of 98.60% using the INbreast dataset and 98.80% using the mini-MIAS dataset. Next, Hekal et al. [5] presented a new computer-aided detection technique for classifying two mammography cancers. The automated optimal Otsu thresholding approach was used in a CAD system to identify tumor-like regions. Then, deep CNNs were used to investigate the AlexNet and ResNet50 architectures, which processed the retrieved TLRs to extract the relevant mammography features. The experiment was acted upon on two datasets, yielding 91% and 84% accuracy rates, respectively. Finally, Siddeeq et al. [29] presented a framework with a ResNet-based customized neural network that was applied to an unbalanced dataset using the data augmentation and pyramid of scales approaches. The results obtained from the mammograms in the INbreast dataset show improved performance when the training dataset is increased.

Hikmah et al. [30] improved the diagnostic outcomes using an image-processing architecture for multi-view screening. The tumor regions were segmented using first-order local entropy, a texture-based method. The feature extraction results were used to calculate the radius and area of likely malignancy. According to the outcomes of this suggested method, the detection accuracy of the CC and MLO views for breast cancer was 88.0% and 80.5%, respectively. Alruwaili et al. [31] presented a framework that emphasized transferable learning. In order to avoid overfitting and achieve reliable results, a variety of augmentation techniques are used to enhance the number of mammograms. Almalki et al. [32] presented a technique on the large mammography images dataset. In the first stage, classification was performed, followed by extracting the pectoral muscle in the next step. In the third stage, abnormal spots in a well-enhanced image were detected using a new segmentation module to recognize breast cancer. The Breast Imaging and Reporting and Data System’s dataset had five categories and obtained an accuracy of 92%.

Moreover, on the Mammographic Image Analysis Society database, the suggested technique yields a result of about 97%. Karthiga et al. [33] presented a methodology combining two main components: transfer learning and CNNs. The hyperparameters of the CNN model were modified to enhance the classification performance. The outcome showed that the presented strategies significantly improved accuracy for the combined datasets (92.27%), for MIAS (95.95%), for DDSM (99.39%), and for INbreast (96.53%). Several other methods were also introduced to classify breast cancer [34]. Few other studies also existed in the literature that proposed deep learning-based frameworks for breast cancer classification, such as the optimized stacking learning approach [35], fuzzy c-mean and median support value-based CNN approach [36], and named a few more [37].

The above studies focused on the model’s information fusion, selection of manual hyperparameter values, data augmentation, and tumor identification using thresholding and CNN techniques. However, it was observed that they missed several important steps that could help improve accuracy. Those steps are contrast enhancement and the optimization of the extracted features. In the deep model, the SGD and ADAM optimizers are normally employed for the weight optimizations. However, we also included a feature optimization technique after the feature extraction process to overcome the computational time, problem of overfitting, and improvement in accuracy.

## 3. Proposed Novel Framework

This section presents the proposed framework for classifying breast cancer using mammogram images. Figure 1 highlights the architecture of this cancer classification. According to this figure, for the experimentation, datasets from INbreast and CBIS-DDSM are utilized. In the first phase, a contrast enhancement technique is employed on both datasets and then data augmentation is performed. A modified EfficientNet-b0 model is employed for training on both the original and improved datasets. For training purposes, a deep transfer learning technique is used. After that, feature extraction is performed from the average pool layer. Then, a hybrid optimization algorithm is employed to select the best features. By using a serial-based approach, the best-selected features are fused and classified using machine learning classifiers.

### 3.1. Datasets

In this section, the details of selected datasets have been presented. Two datasets have been employed for the experimentation of this work, such as the CBIS-DDSM (the Curated Breast Imaging Subset of DDSM), where the digital database is a revised and standardized version of scanning for breast cancer (DDSM) and INbreast. The DDSM dataset comprises two classes, benign and malignant, as shown in Figure 2.

The INbreast dataset comprises two classes that are benign and malignant. Figure 3 illustrates the sample images of this dataset, and it contains 410 images of 115 patients. The size of the images is 2560 × 3328 and 3328 × 4084 pixels. For the experiment, 108 mass mammogram images were employed.

### 3.2. Novelty 1: Contrast Enhancement and Augmentation

Traditional haze-removal techniques aim to produce a high-quality rebuilt image by adjusting the contrast and saturation. The visibility of the scene in the image can be made substantially better by using the haze reduction procedure. In this work, we proposed a new hybrid contrast enhancement technique based on haze removal and local-global transformation. Mathematically, this technique is described as follows:

Consider Δ as an entire image database having N number of images. Let $I\left(x,y\right)$ be an original image of dimensional N×M×3 and $\tilde{CF}\left(x,y\right)$ is the final enhanced image. In the first step, we employed a haze reduction technique based on the dark channel and applied it to the original image. Mathematically, the haze reduction process is defined as follows:(1)$$H\left(x\right)=Y\left(x\right)t\left(x\right)+L\left(1-t\left(x\right)\right)$$
where H denotes the observed intensity value, Y denotes the scene radiance, $t\left(x\right)$ denotes the transmission map, and L denotes the atmospheric light. The employed dehazing algorithm recovers the scene radiance Y from the estimation of the transmission map and atmospheric light as follows:(2)$$Y\left(x\right)=\frac{\left(H\left(x\right)-\alpha \right)}{\left(\max(t\left(x\right),\;t_{0}\right)}+\alpha$$

The resultant $Y\left(x\right)$ is later utilized to obtain the global contrast of an image based on the following formulation:(3)$$g_{0}=\left(1+C_{k}\right)\times \left(g_{i}-k_{mean}\right)+\sigma$$
where $g_{0}$ is a resultant global contrast image, $C_{k}$ denotes the global contrast gain factor, $g_{i}$ denotes the input pixel value of $Y\left(x\right)$, $k_{mean}$ is a global mean value of $Y\left(x\right),$ and σ denotes the standard deviation of $Y\left(x\right)$. In the latter step, we computed the local contrast of the haze reduction image by employing the following mathematical function:(4)$$L\left(x,y\right)=\phi \left(x,y\right)+\frac{LC}{\sigma \left(i,j\right)+\alpha }\times \left[\phi \left(x,y\right)-\mu \left(x,y\right)\right]$$
where $\phi \left(x,y\right)$ denotes the grayscale pixel of the dehazed image $Y\left(x\right)$, LC denotes local contrast, α denotes a small parameter value, and $\mu \left(x,y\right)$ denotes the mean value of the dehazed image, respectively. Finally, we fused the local and global contrast resultant images into a single image by employing the following mathematical equation and obtaining the final enhanced image.
(5)$$\tilde{CF}\left(x,y\right)=\left[g\left(x,y\right)+L\left(x,y\right)\right]-I\left(x,y\right)$$

The final enhanced image is $\tilde{CF}\left(x,y\right)$, which is further utilized for the augmentation process. Visually, these enhanced images are shown in Figure 4.

### 3.3. Data Augmentation

The small number of image datasets is useful for training traditional machine learning techniques, such as shape features (HOG), point features, color features, and more. For deep learning models, it is always essential to generate or collect some larger datasets. However, the publicly available datasets for breast cancer are not large enough; therefore, we performed data augmentation in this work. In addition to growing the dataset, data augmentation reduces the overfitting issues and strengthens the deep learning model’s robustness. Eight more photos for each identified patch were created by rotating each image four times at the angles of 0°, 90°, 180°, and 270°, and then by flipping the resulting four images from left to right. After the process of augmentation, the summary of images is presented in Table 1. Moreover, Figure 5 displays the sample images.

### 3.4. EfficientNet-B0 Pre-Trained Model

EfficientNet uses a compound coefficient to scale both the design and sizing of its convolutional neural network. This compound coefficient uniformly scales the depth, width, and resolution parameters to maintain consistency throughout the network. Unlike the traditional approach of unconstrained scaling of these parameters, the EfficientNet scaling approach employs a fixed set of scaling coefficients to uniformly adjust the resolution, width, and depth of the network. The concept underlying the compound scaling method is that, when the input images become larger, the network requires additional layers to expand its receptive field and more channels to detect finer details within the larger image. The fundamental EfficientNet-b0 network is composed of squeeze-and-excitation blocks, as well as MobileNetV2 inverted bottleneck residual blocks. The architecture of the EfficientNet-b0 model is shown in Figure 6.

**Fine-Tuned Model:** Originally, the EfficientNet-b0 network was trained using more than a million images of the ImageNet dataset that contains 1000 different object categories. For fine-tuning, we initially excluded the last three layers and added three new layers, such as the fully connected, softmax, and output layers. After that, we initialized some hyperparameters, such as a learning rate of 0.005, momentum of 0.703, epochs of 100, and the stochastic gradient descent as an optimizer. Finally, we trained this fine-tuned model using deep transfer learning.

Knowledge transfer from one field or domain to another is called transfer learning. Deep learning involves a great deal of training data, making it difficult and time-consuming to learn specific patterns, especially in medical imaging. A labeled dataset serves as the “source,” which is indicated as:(6)$$F_{\$}=\left(G_{\$},H_{\$}\right)$$
where $H_{\$}\in I^{r_{1}}$ is the label, and $G_{\$}\in I^{\partial Xr_{1}}$ is a d-dimensional feature space. The $r_{1}$ is the total number of source samples. The “target” dataset is identified as $F_{\tau =}\left(G_{\tau }\right)$, where $G_{\tau }\epsilon I^{\partial Xr_{2}}$ and $r_{2}$ are the total number of target samples. There are two important definitions as follows:

The two primary components of a “domain” $F=\left\{G,\rho \left(i\right)\right\}$ are the feature data G and the associated marginal probability distribution $\rho \left(i\right)$, which has a range of [0, 1]. The (Task) can be defined as labels H and their matching function $J\left(i\right)$, which anticipates them, making up a “task” $\tau =\left\{H,\;J\left(i\right)\right\}$. Overall, this process is shown in Figure 7.

After the training of the deep learning models using deep transfer learning, deep features were extracted. Deep features were extracted for both datasets, such as CBIS-DDSM and INbreast. As shown in Figure 1, the training of the fine-tuned model is conducted separately on original and enhanced images. The features were extracted from each trained model’s global average pool layers. After the feature extraction, we obtained feature vectors of dimensions N×1280 for the CBIS-DDSM dataset (two vectors of the same dataset) and N×1280 for the INbreast dataset (two vectors of the same dataset).

### 3.5. Novelty 2: Binomial Probability Serially Fusion

Consider that we have two feature vectors $\partial_{1}$ and $\partial_{2}$ of CBIS-DDSM, which have the dimensions N×1280 and N×1280. Similarly, we have two feature vectors ${\overset{ˇ}{\partial }}_{1}$ and ${\overset{ˇ}{\partial }}_{2}$ of the INbreast dataset, with the dimensions N×1280 and N×1280, respectively. Suppose that ${\overset{ˇ}{\partial }}_{x}$ and ${\overset{ˇ}{\partial }}_{x1}$ denote the fused feature vectors using a simple serial-based approach and are mathematically defined as follows:(7)$${\overset{ˇ}{\partial }}_{x}={\left(\begin{array}{c} \partial_{1}\\ \partial_{2} \end{array}\right)}_{\left(N\times 1280+N\times 1280\right)}$$
(8)$${\overset{ˇ}{\partial }}_{x1}={\left(\begin{array}{c} {\overset{ˇ}{\partial }}_{1}\\ {\overset{ˇ}{\partial }}_{2} \end{array}\right)}_{\left(N\times 1280+N\times 1280\right)}$$

Both fused vectors ${\overset{ˇ}{\partial }}_{x}$ and ${\overset{ˇ}{\partial }}_{x1}$ are analyzed and further refined using the proposed serially extended mean deviation approach. In this approach, the binomial probability distribution is initially computed for the entire feature vector and then defined as a final activation function based on the resultant value used for the fusion process. Mathematically, this process is defined as follows:(9)$$P\left({\overset{ˇ}{\partial }}_{x}\right)=nC_{{\overset{ˇ}{\partial }}_{x}}p^{{\overset{ˇ}{\partial }}_{x}}{\left(1-p\right)}^{n-{\overset{ˇ}{\partial }}_{x}}$$
(10)$$P\left({\overset{ˇ}{\partial }}_{x1}\right)=nC_{{\overset{ˇ}{\partial }}_{x1}}p^{{\overset{ˇ}{\partial }}_{x1}}{\left(1-p\right)}^{n-{\overset{ˇ}{\partial }}_{x1}}$$
Using $P\left({\overset{ˇ}{\partial }}_{x}\right)$ and $P\left({\overset{ˇ}{\partial }}_{x1}\right)$, the final activation functions have been defined for the final fusion.
(11)$$Act1=\left\{\begin{array}{ll} FU1\;\;\;\;\;\;\;\;for & {\overset{ˇ}{\partial }}_{x}\geq P\left({\overset{ˇ}{\partial }}_{x}\right)\\ Ignore & Elsewhere \end{array}\right.$$
(12)$$Act2=\left\{\begin{array}{ll} FU2\;\;\;\;\;\;\;\;for & {\overset{ˇ}{\partial }}_{x1}\geq P\left({\overset{ˇ}{\partial }}_{x1}\right)\\ Ignore & Elsewhere \end{array}\right.$$
where FU1 and FU2 denote the fused feature vectors of the CBIS-DDSM and INbreast datasets, respectively. In our work, the dimensions of these feature vectors were N×1726 and N×1702, respectively. Furthermore, the resultant feature vectors were optimized further using the proposed Equilibrium-Jaya controlled Regula Falsi method.

### 3.6. Novelty 3: Proposed Features Selection Method

This work selects the best features by employing a hybrid optimization algorithm named Equilibrium-Jaya controlled Regula Falsi. The main purpose of the hybridization of both algorithms is to obtain the most optimal features that can help in improved accuracy and a reduction in computational time.

#### Equilibrium Optimizer Algorithm

Initialization: In this stage, the equilibrium optimizer employs a collection of particles, each representing a concentration vector that holds the answer to the optimization issue. By using the following formula, a random initial concentrations vector is created in the search space [41]:(13)$$\vec{x_{j}}=u_{min}+\left(u_{max}-u_{min}\right)*v\;\;\;\;\;\;\;\;\;\;\;\;\;\;j=1,2,3\ldots ,n$$
where $\vec{x_{j}}$ represents the particle n’s concentration vector and $u_{min},$ $u_{max}$ set the upper and lower bounds, respectively. For each dimension in the challenge, v is an arbitrary number in the range of [0, 1], and n indicates the number of particles present.

Equilibrium Pool and Candidates: Each meta-heuristic algorithm attempts to attain a goal based on its nature. For instance, WOA looks for predators. An artificial bee colony (ABC) looks for a food supply, and in relation to EO, it looks for the system’s equilibrium state. The near-optimal solution to the optimization issue may be reached by EO when it reaches the equilibrium state. During optimization, EO is unaware of the degree of levels of concentration that reach equilibrium. The four best particles identified in the community at equilibrium have members assigned, along with another candidate with the average of the four best particles. These five equilibrium members assist EO in its role as an exploitation and exploration operator, with the first four members assisting EO in improving its capacity for diversification and exploitation on average. These five candidates are kept in an equilibrium pool vector as follows [41]:(14)$${\vec{\omega }}_{Eq,pool}=\left[{\vec{\omega }}_{Eq\left(1\right)},{\vec{\omega }}_{Eq\left(2\right)},{\vec{\omega }}_{Eq\left(3\right)},{\vec{\omega }}_{Eq\left(4\right)},{\vec{\omega }}_{Eq\left(Avg\right)}\right]$$

Updating the Concentration: Using the following phrase aids EO in maintaining a tenable balance between intensification and diversity. Since the turnover rate in a real control volume can change over time, k is meant to be a random vector between 0 and 1.
(15)$$\vec{H}=e^{-\vec{\delta }\left(IT-IT_{0}\right)}$$
(16)$$IT={\left(1-\frac{Iter}{IT_{max}}\right)}^{\left(\alpha 2*\left(\frac{Iter}{IT_{max}}\right)\right)}$$
where the current and maximum iterations are denoted by Iter and $IT_{max}$, respectively. Additionally, the exploitation capability is managed by the constant variable α2. Another factor is α1, which is employed to enhance the diversification and intensity of EO:(17)$${\vec{IT}}_{0}=\frac{1}{\vec{\delta }}\ln\left(-\alpha 1\;Sign\;\left(\vec{v}-0.5\right)\;[1-e^{-\vec{\delta }IT}\right)+IT$$
where α1 is a constant used to control the exploration capabilities; as α1 increases, diversification improves while the intensity decreases. Unlike α1, α2 is a fixed value utilized to regulate the exploitation capacity. The intensification capability is good, and the diversification capability is worse when α2 is higher. Another term that is used to increase the intensification operator is the generation rate (GR), which is defined as follows [41]:(18)$$\vec{GR}={\vec{GR}}_{0}*e^{-\vec{\delta }*\left(IT-IT_{0}\right)}$$
where, $GR_{0}$ is the beginning value, and δ is an arbitrary vector with a range from 0 to 1:(19)$${\vec{GR}}_{0}=\vec{GRC}*\left(\vec{u_{Eq}}-\vec{\delta }*\vec{U}\right)$$
(20)$$\vec{GRC}=\left\{\begin{array}{ll} 0.65v_{1} & v_{2}>RC\\ 0 & Otherwise \end{array}\right.$$
where the random values $v_{1}$ and $v_{2}$ range from 0 to 1. The generation rate control parameter in this equation is denoted by $\vec{GRC}$, that, based on a probability, RC determines whether the generation rate will be implemented in the updating process. Last but not least, the EO updating equation is as follows:(21)$$\vec{U}=\vec{u_{Eq}}+\left(\vec{U}-\vec{u_{Eq}}\right)+\vec{H}+\frac{\vec{GR}}{\vec{\delta }*x}\left(1-\vec{H}\right)$$

The output of this algorithm $\vec{U}$ is again analyzed and modified with the Jaya algorithm. The working principle of the algorithm is that the seeking agents or particles continuously try to approach the goal by ignoring the worst solution in each iteration. By retaining the best position and completely discarding all other positions, the seeking agent updates its position based on this method. As a result, all solutions produced by iteration are better than the prior worst solution. Every suggested solution or searching agent in JA is known as a particle $V_{\sigma ,\emptyset ,\rho }$. Each particle searches for the best and optimal solution and avoids the worst solution of the objective or cost function in the search region. With “a” candidates $\left(i.e.,\;\sigma =1,\;2,\ldots ,\;a\right)$ and “b” design variables $\left(i.e.,\;\emptyset =1,\;2,\ldots ,\;b\right),$ this is accomplished in order to maximize the objective function (J). The particle locations are mathematically updated as follows:(22)$${V^{\prime }}_{\sigma ,\emptyset ,\rho }=V_{\sigma ,\emptyset ,\rho }+z_{1,\sigma ,\rho }\left(V_{\sigma ,best,\rho }-\left|V_{\sigma ,\emptyset ,\rho }\right|\right)-z_{2,\sigma ,\rho }\left(V_{\sigma ,worst,\rho }-\left|V_{\sigma ,\emptyset ,\rho }\right|\right)$$
where $V_{q,r,s}$ represents the ∅th candidate’s σth variable during the ρth iteration. Random numbers with values between [0, 1] are $z_{1,\sigma ,\rho }$ and $z_{2,\sigma ,\rho }$. The $V_{\sigma ,best,\rho }$ and $V_{\sigma ,worst,\rho }$, respectively, are the best and worst candidate values. The updated value of $V_{\sigma ,\emptyset ,\rho }$ is ${V^{\prime }}_{\sigma ,\emptyset ,\rho }$. If the value returned by $V_{\sigma ,\emptyset ,\rho }$ is better than ${V^{\prime }}_{\sigma ,\emptyset ,\rho }$ then the ${V^{\prime }}_{\sigma ,\emptyset ,\rho }$ value is retained. The cost function is employed in each iteration, and the resultant features are used to compute the best value using the Regula Falsi method. Mathematically, the Regula Falsi method is defined as follows:(23)$$RF=x_{0}-\left[f\left(x_{0}\right)\times \frac{x_{1}-x_{0}}{f\left(x_{1}\right)-f\left(x_{0}\right)}\right]$$
(24)$$RF_{n}=x_{n-2}-\left[f\left(x_{n-2}\right)\times \frac{x_{n-1}-x_{n-2}}{f\left(x_{n-1}\right)-f\left(x_{n-2}\right)}\right]$$

The output of this function is passed to the fitness function to check the accuracy. The fine KNN is employed in this work as a function. The selected features, after this process, are finally passed to the classifiers for classification. The dimensions of the proposed selected features are N×826 and N×812, which was previously N×1726 (the fusion of dataset 1, and 1702 for the fusion of dataset 2). This shows that the proposed technique significantly reduced the size of the feature vector.

The reasons for feature selection (FS): The aim of feature selection is to reduce the problem of overfitting and the overall computational cost. In this work, the purpose of the proposed FS technique is to choose the most optimal features that maintain the classification accuracy or improve the accuracy but significantly reduce the computational time. We compared the proposed FS technique with other techniques, such as the principal component analysis (PCA), genetic algorithm (GA), and PSO. In PCA, the features are reduced as per the experience, but in the GA and PSO, 20–30% of the features are reduced, and 70% (maximum) of the features are selected (in our experiments).

## 4. Results

The proposed architecture results for breast cancer classification have been presented in this section in terms of numeric values, confusion matrix, and plots. The augmented CBIS-DDSM and INbreast publicly available datasets have been employed for the experimental process (details of the datasets have been given under Section 3.1). The training and testing ratio is defined as 50: 50, with a cross-validation value of 10. Several classifiers have been utilized for classification, such as ensemble subspace KNN, fine KNN, cubic SVM, medium Gaussian SVM, medium neural networks, wide neural networks, and weighted KNN, to name but a few. The outcomes are calculated using several experiments: (i) a classification using deep features on the original dataset; (ii) a classification using deep features on an enhanced dataset; (iii) a fusion of the original and improved dataset deep features using the proposed fusion technique; and (iv) a feature selection using the proposed Equilibrium-Jaya controlled Regula Falsi method. The entire experimental process was conducted on MATLAB 2022a using a desktop computer with 16 GB of RAM and an 8 GB graphics card.

### 4.1. CBIS-DDSM Results

Table 2 describes the outcomes of the classification of the CBIS-DDSM dataset using the original dataset’s deep features. The ensemble subspace KNN classifier obtained the best-obtained accuracy of 92%. The sensitivity rate of this classifier is 92.05, the precision rate is 92.05, the F1 score is 92.05, and the FNR is 7.95%, respectively. The second best-obtained accuracy is 91.8%, achieved by fine KNN. The newly added classifiers, such as the narrow neural network (**N^3^**), medium NN, wide NN, and bi-layered NN, obtained 87.5, 88.7, 88.4, and 87.2% accuracy, respectively. The computational time of each classifier is listed in this table. Based on the mentioned time, the fine KNN classifier execution is less than the other classifiers, such as 50.021 (s); however, the maximum noted time for the ensemble subspace KNN classifier is 584.69 (s).

Table 3 describes the outcomes of the classification of the CBIS-DDSM dataset after the proposed contrast enhancement technique. The best-obtained accuracy of 95% was achieved by the ensemble subspace KNN classifier. The sensitivity rate of this classifier is 95.05, the precision rate is 95, the F1 score is 95.02, and the FNR is 4.95%. The second best-obtained accuracy is 94.7%, achieved by fine KNN. The noted accuracy for N^3^ is 90.1, the medium NN is 90.6, the wide NN is 90.5, and the bi-layered NN is 89.9%, respectively. Comparing these obtained accuracies after the proposed contrast enhancement step shows that the performance is more improved than the original dataset, as shown in Table 2. Computationally, the fine KNN classifier execution is less than the other classifiers, such as 54.658 (s); however, the maximum noted time is 633.3 (s) for the ensemble subspace KNN. Compared to Table 2, it is observed that the accuracy of this experiment is improved, which shows the strength of this classifier.

Table 4 describes the outcomes of the classification of the CBIS-DDSM dataset using the proposed feature fusion approach. The best-obtained accuracy after the feature fusion is 94.1% by ensemble subspace KNN. The sensitivity rate of this classifier is 94.15, the precision rate is 94.15, the F1 score is 94.15, and the FNR is 5.85%. The second best-obtained accuracy is 93.8%, achieved by fine KNN. The accuracy is also computed for several neural networks, such as N^3^, MNN, WNN, and Bi-NN, which are 90.3, 92.2, 92.2, and 90.5%, respectively. A comparison of these values with Table 2 shows that the accuracy is enhanced after the fusion process. Figure 8 illustrates the confusion matrix of the ESKNN classifier. This figure shows that the malignant class has a correct prediction rate of 93.4%. Moreover, Figure 9 displays that the computational time after the fusion process is increased, which is a drawback of this step; thus, this is why we proposed a feature selection technique.

Table 5 describes the outcomes of the classification of the CBIS-DDSM dataset using the proposed feature selection algorithm. The best-obtained accuracy of 95.4%, achieved by ensemble subspace KNN, is more improved than the originally extracted features, enhanced image features, and proposed fusion. The sensitivity rate of this classifier is 95.4, the precision rate is 95.35, the F1 score is 95.37, and the FNR is 4.6%. The classification accuracy for the neural networks is also improved after the proposed selection method, such as N^3^ is 94.1, MNN is 94.7, WNN is 90.7, and Bi-NN is 94.9%, respectively. Figure 10 shows the confusion matrix of the ESKNN classifier that shows the correct predicted values in a diagonal. The computational time is also reduced after the proposed selection method and is plotted in Figure 11. Overall, the accuracy is enhanced after the proposed feature selection method. Moreover, time is also reduced when compared to the previous experiments.

### 4.2. INbreast Dataset Results

The outcomes of the classification of the INbreast cancer dataset have been discussed in this subsection. Table 6 describes the outcomes of the classification of the INbreast dataset using the original images dataset. Deep features are extracted from the deep model and obtained the maximum accuracy of 98.3% on medium Gaussian SVM (MGSVM). The sensitivity rate of this classifier is 98.25, the precision rate is 98.25, the F1 score is 98.25, and the FNR is 1.75%. The computational time is noted for each classifier, listed in this table. Based on the mentioned time, the quadratic SVM (QSVM) classifier execution is less than the other classifiers.

Table 7 describes the outcomes of the classification of the INbreast dataset using the proposed enhanced images. After the training on enhanced images, we obtained a testing accuracy of 98.1% on MGSVM. The sensitivity rate of MGSVM for this experiment is 98.05, the precision rate is 98.15, the F1 score is 98.09, and the FNR is 1.95%, respectively. The computational time is noted for each classifier, listed in this table. However, it was discovered that the accuracy remains consistent after the enhancement, but the computational time is reduced.

Table 8 displays the outcomes of the proposed feature fusion for the INbreast dataset. After the proposed fusion, we obtained an accuracy of 99.6% on MGSVM. The sensitivity rate of this classifier is 99.55, the precision rate is 99.6, the F1 score is 99.57, and the FNR is 0.45%. These values can be further calculated through a confusion matrix, given in Figure 12. In this figure, the diagonal values display the correct or accurate predicted values. The computational time is noted for each classifier, and it is observed that the time is increased after the proposed fusion process. However, the accuracy is significantly increased, which is the strength of this step. We proposed a feature selection approach to resolve the high computational time problem.

The outcomes of the proposed features are presented in Table 9 and achieved the maximum accuracy of 99.4% on cubic SVM. The sensitivity rate of this classifier is 99.4, the precision rate is 99.4, the F1 score is 99.4, and the FNR is 0.6%. A confusion matrix is illustrated in Figure 13, in which the diagonal values show the correct prediction. The computational time is noted for each classifier, listed in this table, and the noted minimum time is 2.1975 for the cubic SVM classifier. Compared to the original image features, contrast-enhanced image features, and fused features, the proposed selection method shows improved performance for both accuracy and time. The visual comparison, in terms of the computational time, is shown in Figure 14, which shows the clear strength of the proposed feature selection.

Finally, we compare the proposed framework with some state-of-the-art techniques which used similar datasets. Table 10 compares the proposed framework with the recent techniques, and it is noted that Surendiren et al. [42] obtained an accuracy of 93.3% on the CBIS-DDSM dataset. Houbey et al. [43] obtained an improved accuracy of 96.52% on the INbreast dataset. In [44], the authors obtained 85.38% and 99% accuracy for the CBIS-DDSM and INbreast datasets. This work achieves an accuracy of 95.4% for CBIS-DDSM and 99.7% for the INbreast cancer dataset, which shows an improvement.

## 5. Conclusions

This work proposes a novel framework for breast cancer classification from mammography images. The proposed framework comprises important steps, starting with image acquisition and classification. In the first phase, a contrast enhancement technique is proposed. The resultant enhanced images have been used to train the deep learning model (EfficientNet-b0) and compare the results with the original image’s deep feature accuracy. The results show that the accuracy of the proposed enhancement technique is better, but the recently obtained accuracy was not met; therefore, a new fusion technique is proposed. The original image and enhanced image features have been fused using the proposed fusion technique and thus show a significant improvement in accuracy. The drawback of this step was the increase in computational time; therefore, a new feature selection technique is proposed, called Equilibrium-Jaya controlled Regula Falsi. After employing the proposed selection technique, time is significantly reduced for both datasets.

### Limitations and Future Directions

The limitation of this work is the fusion process that consumes much time compared to the feature selection technique. In addition, the manual initialization of the hyperparameters is not an efficient method. These limitations will be resolved in the future by employing the following steps:-The tumor segmentation step will be added using image fusion techniques [49], and later-stage segmented regions will be considered for the feature extraction.-An optimized feature fusion technique will be considered for resolving the computational time problem.-The Bayesian optimization technique will have opted for the hyperparameters’ initialization.

## Figures and Tables

**Figure 1 diagnostics-13-01238-f001:**
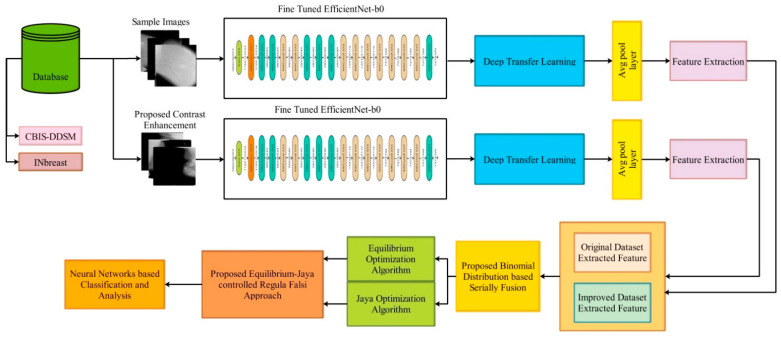
The proposed framework of an optimal deep learning feature fusion of the classification of breast cancer.

**Figure 2 diagnostics-13-01238-f002:**
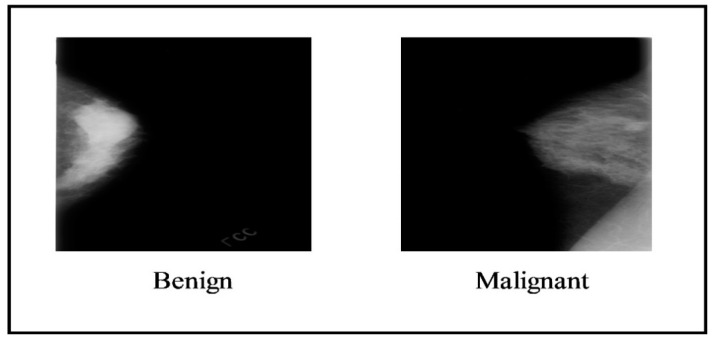
Sample images of the CBIS-DDSM dataset [38].

**Figure 3 diagnostics-13-01238-f003:**
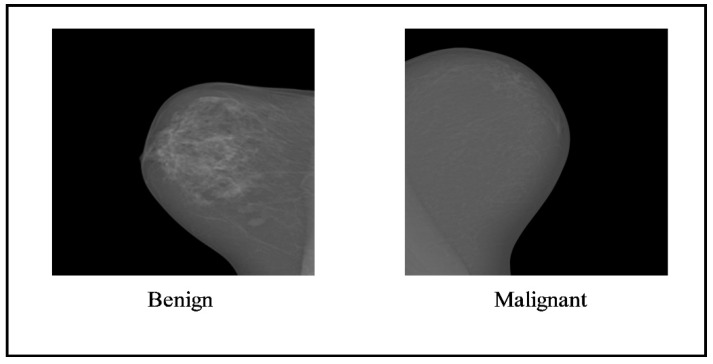
Sample images of the INbreast dataset [39].

**Figure 4 diagnostics-13-01238-f004:**
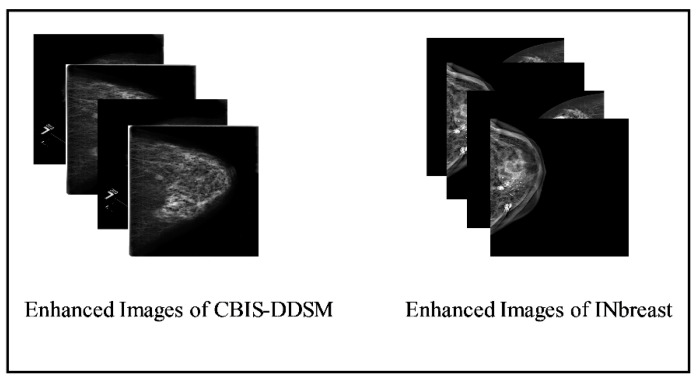
Visual illustration of proposed enhancement technique.

**Figure 5 diagnostics-13-01238-f005:**
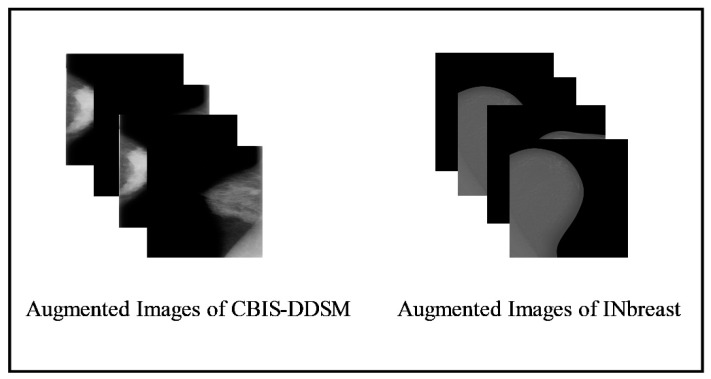
Visual illustration of data augmentation step.

**Figure 6 diagnostics-13-01238-f006:**
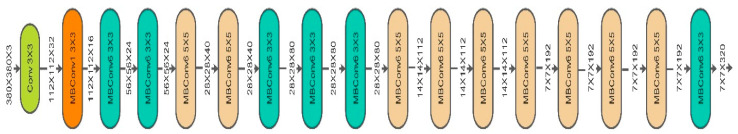
The architecture of EfficientNet-b0 [40].

**Figure 7 diagnostics-13-01238-f007:**
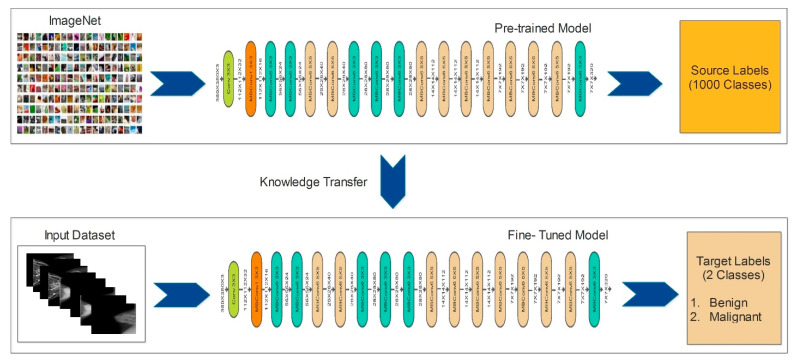
Visual process of deep transfer learning for breast cancer classification.

**Figure 8 diagnostics-13-01238-f008:**
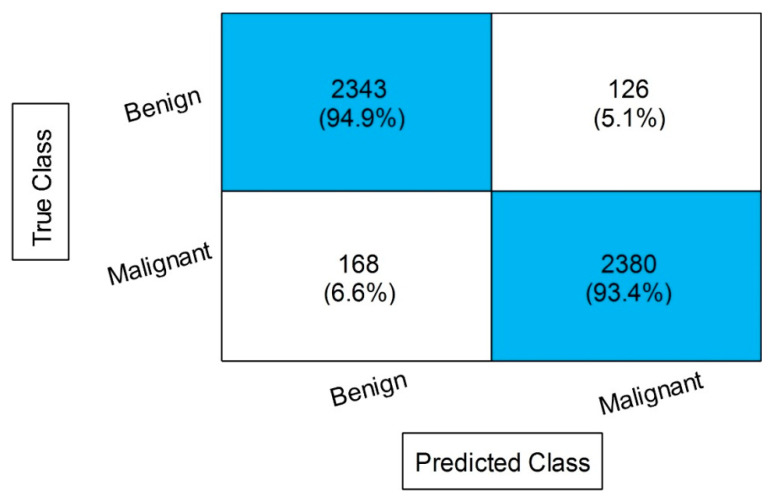
Confusion matrix of ESKNN after the proposed feature fusion.

**Figure 9 diagnostics-13-01238-f009:**
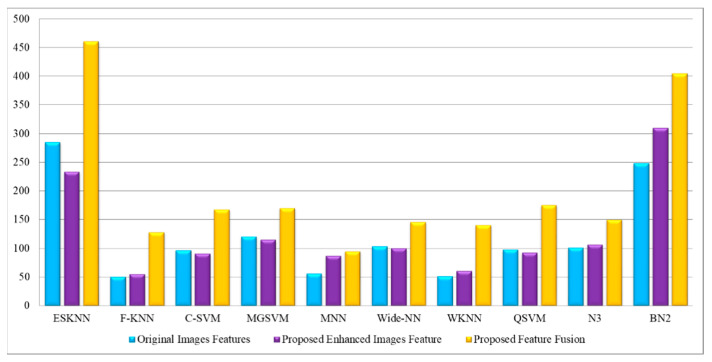
Comparison of proposed fusion step in terms of computational time with Table 2 and Table 3.

**Figure 10 diagnostics-13-01238-f010:**
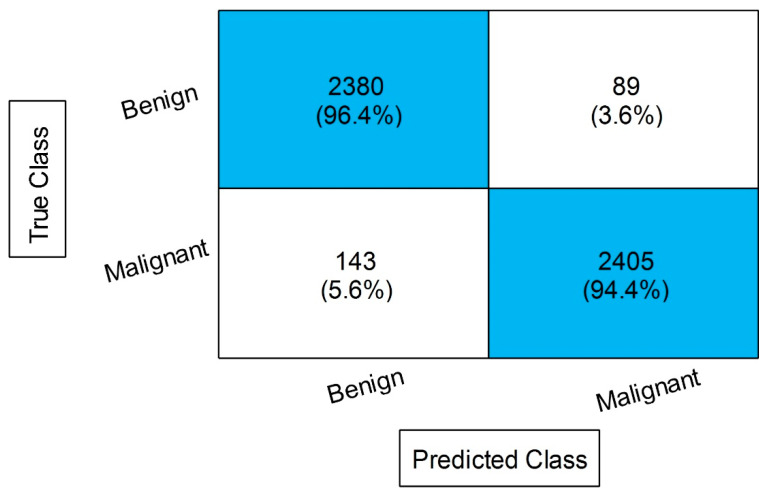
Confusion matrix of ESKNN classifier using proposed feature selection method.

**Figure 11 diagnostics-13-01238-f011:**
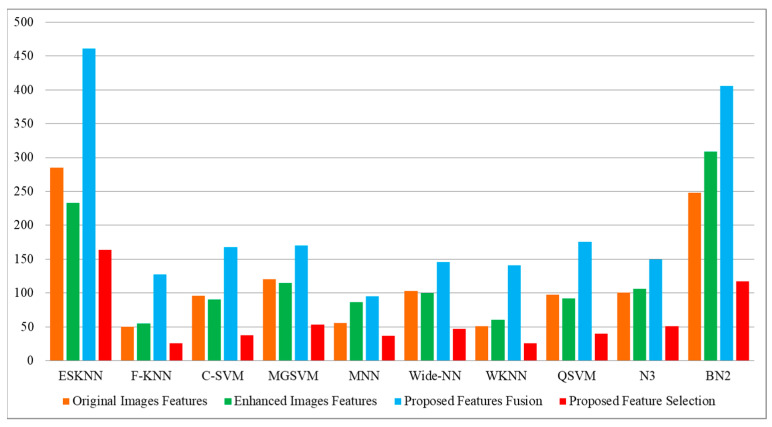
Comparison of proposed feature selection technique in terms of computational time with Table 2, Table 3 and Table 4.

**Figure 12 diagnostics-13-01238-f012:**
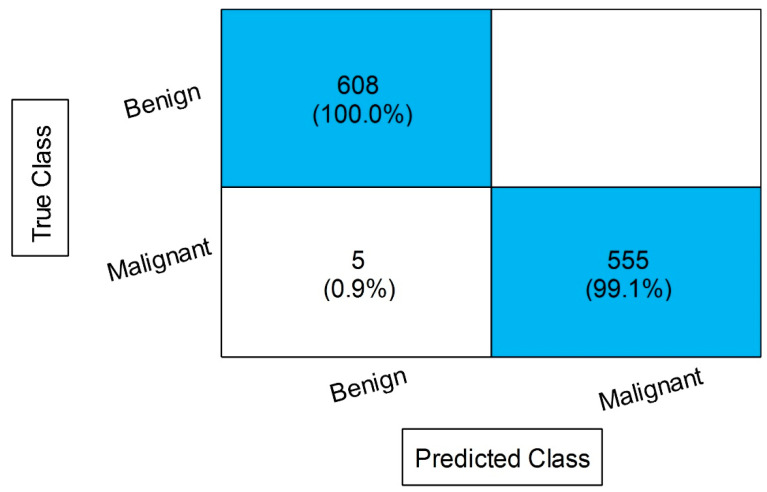
Confusion matrix MGSVM after employing proposed feature fusion.

**Figure 13 diagnostics-13-01238-f013:**
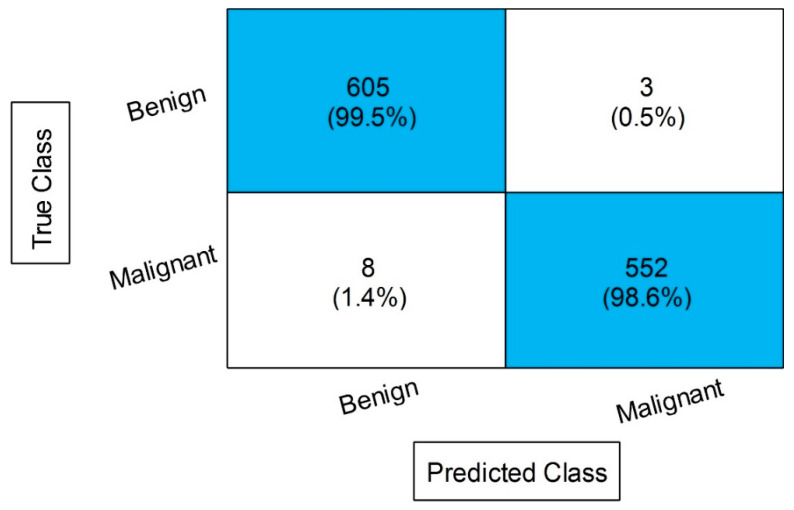
Confusion matrix of CSVM after employing the proposed feature selection.

**Figure 14 diagnostics-13-01238-f014:**
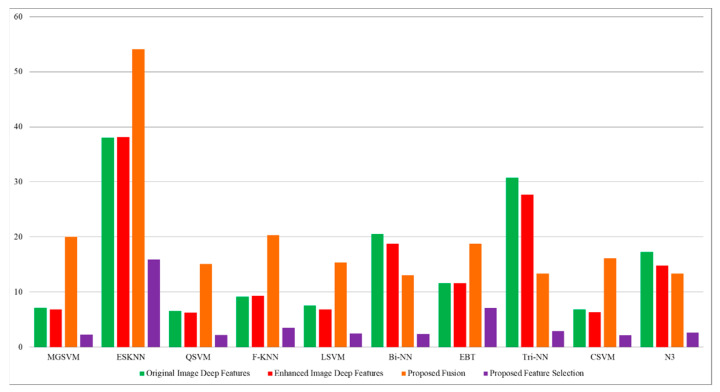
Computational time-based visual comparison among all four experiments.

**Table 1 diagnostics-13-01238-t001:** Summary of augmented and original datasets.

**CBIS-DDSM**			
**Original Dataset**		**Augmented Dataset**	
**Class**	**# Images**	**Class**	**# Images**
Benign	557	Benign	4939
Malignant	637	Malignant	5096
**INbreast Dataset**			
**Original Dataset**		**Augmented Dataset**	
**Class**	**# Images**	**Class**	**# Images**
Benign	76	Benign	1216
Malignant	70	Malignant	1120

**Table 2 diagnostics-13-01238-t002:** Classification results of original dataset after deep feature extraction using fine-tuned EfficientNet-b0. * ESKNN denotes the ensemble subspace KNN, F-KNN denotes fine KNN, C-SVM denotes cubic SVM, MGSVM denotes medium Gaussian SVM, MNN denotes the medium neural network, WNN denotes wide neural network, N^3^ denotes the narrow neural network, and BN^2^ denotes the bi-layered neural network.

* Classifier	Sensitivity Rate %	Precision Rate %	F1 Score	AUC	Accuracy %	FNR	Time (s)
ESKNN	92.05	92.05	92.05	0.96	92	7.95	584.69
F-KNN	91.9	91.85	91.87	0.92	91.8	8.1	50.021
C-SVM	90.35	90.35	90.35	0.96	90.4	9.65	95.967
MGSVM	88.7	88.7	88.7	0.95	88.7	11.3	120.35
MNN	88.7	88.65	88.67	0.93	88.7	11.3	55.385
Wide NN	88.4	88.45	88.42	0.94	88.4	11.6	103.12
WKNN	88.1	88.1	88.1	0.95	88.1	11.9	51.004
QSVM	87.6	87.6	87.6	0.95	87.6	12.4	97.295
N^3^	87.55	87.55	87.55	0.92	87.5	12.45	100.66
BN^2^	87.25	87.25	87.25	0.92	87.2	12.75	247.92

**Table 3 diagnostics-13-01238-t003:** Classification results after employing the proposed contrast enhancement step for deep features extraction.

Classifier	Sensitivity Rate (%)	Precision Rate (%)	F1 Score (%)	AUC	Accuracy (%)	FNR (%)	Time Complexity (s)
ESKNN	95.05	95	95.02	0.97	95.0	4.95	633.3
F-KNN	94.7	94.7	94.7	0.95	94.7	5.3	54.658
C-SVM	92.15	92.15	92.15	0.97	92.1	7.85	90.261
MGSVM	91.5	91.5	91.5	0.97	91.5	8.5	114.64
MNN	90.55	90.55	90.55	0.94	90.6	9.45	86.577
Wide NN	90.5	90.5	90.5	0.96	90.5	9.5	99.719
WKNN	90	90	90	0.96	89.9	10	60.308
QSVM	90.4	90.45	90.42	0.96	90.4	9.6	92.179
N^3^	90.1	90.15	90.12	0.94	90.1	9.9	105.76
BN^2^	89.85	89.85	89.85	0.94	89.9	10.15	308.88

**Table 4 diagnostics-13-01238-t004:** Classification results of CBIS-DDSM using proposed feature fusion approach. The bold represent the significant value.

Classifier	Sensitivity Rate %	Precision Rate %	F1 Score	AUC	Accuracy %	FNR	Time Complexity (sec)
ESKNN	94.15	94.15	94.15	0.97	**94.1**	5.85	961.1
F-KNN	93.75	93.75	93.75	0.94	93.8	6.25	**127.76**
C-SVM	93.6	93.6	93.6	0.99	93.6	6.4	167.34
MGSVM	93.1	93.1	93.1	0.98	93.1	6.9	170.19
MNN	92.2	92.25	92.22	0.98	92.2	7.8	94.742
Wide NN	92.1	92.1	92.1	0.98	92.1	7.9	145.807
WKNN	91.75	91.75	91.75	0.97	91.7	8.25	140.53
QSVM	93.15	93.2	93.17	0.98	93.2	6.85	175.34
N^3^	90.35	90.3	90.32	0.94	90.3	9.65	149.832
BN^2^	90.55	90.5	90.52	0.94	90.5	9.45	405.4

**Table 5 diagnostics-13-01238-t005:** Classification of CBIS-DDSM using the proposed feature selection algorithm.

Classifier	Sensitivity Rate (%)	Precision Rate (%)	F1 Score (%)	AUC	Accuracy (%)	FNR (%)	Time Complexity (sec)
Classifier	95.4	95.35	95.37	0.98	95.4	4.6	164.06
ESKNN	95.25	95.25	95.25	0.95	95.3	4.75	25.859
F-KNN	91.75	91.75	91.75	0.97	91.8	8.25	37.899
C-SVM	90.75	90.75	90.75	0.96	90.7	9.25	52.97
MGSVM	92.05	92.05	92.05	0.97	92	7.95	36.853
MNN	94.7	94.7	94.7	0.95	94.7	5.3	47.295
Wide NN	90.7	90.65	90.67	0.96	90.7	9.3	**25.815**
WKNN	89.75	89.75	89.75	0.95	89.1	10.25	39.806
QSVM	91.5	91.5	91.5	0.97	91.5	8.5	50.785
N^3^	94.9	94.85	94.87	0.98	94.9	5.1	117.13

**Table 6 diagnostics-13-01238-t006:** Classification results of INbreast dataset using original images based on fine-tuned deep model features.

Classifier	Sensitivity Rate %	Precision Rate %	F1 Score	AUC	Accuracy %	FNR	Time Complexity (sec)
MGSVM	98.25	98.25	98.25	0.99	**98.3**	1.75	7.1611
ESKNN	98.15	98.15	98.15	0.98	98.2	1.85	38.082
QSVM	98.08	98.05	98.06	1	98.1	1.95	**6.6397**
F-KNN	98.05	98.05	98.05	0.98	98.1	1.95	9.1516
LSVM	97.95	97.95	97.95	1	98	2.05	7.5417
Bi-NN	98	98	98	1	98	2	20.558
EBT	97.85	97.85	97.85	0.99	97.9	2.15	11.611
Tri-NN	97.9	97.9	97.9	1	97.9	2.1	30.736
C-SVM	97.7	97.7	97.7	0.99	97.7	2.3	6.837
N^3^	97.7	97.7	97.7	0.99	97.7	2.3	17.319

**Table 7 diagnostics-13-01238-t007:** Classification results of INbreast dataset using proposed enhanced images based on fine-tuned deep model features.

Classifier	Sensitivity Rate %	Precision Rate %	F1 Score	AUC	Accuracy %	FNR	Time Complexity (sec)
MGSVM	98.05	98.15	98.09	0.99	**98.1**	1.95	6.8656
ESKNN	98	98.1	98.04	0.98	98.0	2	38.111
QSVM	97.9	98	97.94	0.99	97.9	2.1	**6.3209**
F-KNN	97.9	98	97.94	0.98	97.9	2.1	9.3578
LSVM	97.25	95.5	96.36	1	97.3	2.75	6.844
Bi-NN	97.35	97.3	97.32	1	97.3	2.65	18.814
EBT	96.45	96.5	96.47	0.99	96.5	3.55	11.588
Tri-NN	97.35	97.35	97.35	0.99	97.3	2.65	27.701
C-SVM	97.85	97.85	97.85	0.99	97.7	2.15	6.3782
N^3^	97.7	97.7	97.7	0.99	97.7	2.3	14.795

**Table 8 diagnostics-13-01238-t008:** Classification results of proposed feature fusion on INbreast dataset.

Classifier	Sensitivity Rate (%)	Precision Rate (%)	F1 Score	AUC	Accuracy (%)	FNR	Time Complexity (sec)
MGSVM	99.55	99.6	99.57	1	99.6	0.45	20.066
ESKNN	97.9	98	97.94	0.99	97.9	2.1	54.093
QSVM	99.45	99.5	99.47	1	99.5	0.55	15.09
F-KNN	97.8	97.9	97.84	0.98	97.9	2.2	20.399
LSVM	99.45	99.5	99.47	1	99.5	0.55	15.352
Bi-NN	99.15	99.2	99.17	1	99.1	0.85	**13.076**
EBT	98.85	98.9	98.87	1	98.9	1.15	18.83
Tri-NN	99.15	99.2	99.17	1	99.1	0.85	13.374
CSVM	99.45	99.5	99.47	1	99.5	0.55	16.174
N^3^	99.3	99.35	99.32	1	99.3	0.7	13.36

**Table 9 diagnostics-13-01238-t009:** Classification results of proposed feature selection technique on INbreast dataset.

Classifier	Sensitivity Rate %	Precision Rate %	F1 Score	AUC	Accuracy %	FNR	Time Complexity (sec)
MGSVM	99.05	99.1	99.07	1	99.1	0.95	2.2698
ESKNN	98	98.1	98.04	1	98.0	2	15.915
QSVM	99.2	99.25	99.22	1	99.2	0.8	2.2299
F-KNN	97.55	97.55	97.55	0.98	97.6	2.45	3.5232
LSVM	99.2	99.2	99.2	1	99.2	0.8	2.51
Bi-NN	98.2	98.2	98.2	1	98.2	1.8	2.4121
EBT	98.2	98.2	98.2	1	98.3	1.8	7.1051
Tri-NN	98.6	98.6	98.6	0.99	98.5	1.4	2.896
CSVM	99.4	99.4	99.4	1	**99.4**	0.6	**2.1975**
N^3^	99.05	99.05	99.05	1	99.1	0.95	2.6468

**Table 10 diagnostics-13-01238-t010:** Comparison of the proposed framework with recent state-of-the-art techniques.

Reference	Year	Datasets	Accuracy
[42]	2015	CBIS-DDSM	93.3%
[45]	2019	CBIS-DDSM	87.2%
[46]	2020	INbreast	90.9
[47]	2020	CBIS-DDSM	93.47
[1]	2021	CBIS-DDSM	94.7
[43]	2021	INbreast	96.52%
[48]	2022	CBIS-DDSM, INbreast	90.68%, 91.28%
[44]	2022	CBIS-DDSM, INbreast	85.38%, 99%
**Proposed**		CBIS-DDSM, INbreast	**95.4%**, **99.7%**

## Data Availability

The datasets used in this work are publicly available.

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
