# Peer review of "BC2NetRF: Breast Cancer Classification from Mammogram Images Using Enhanced Deep Learning Features and Equilibrium-Jaya Controlled Regula Falsi-Based Features Selection"

_diagnostics, 2023, doi:10.3390/diagnostics13071238_

Round 1

Reviewer 1 Report

The manuscript's purpose is to propose a novel framework for breast cancer classification from mammography images. Authors claim to achieve better performance using their proposed technique. To justify their experiment and result, authors need to do a better literature survey, present the experiment better, explain the rationale behind choices (the techniques), and work on the result section by bringing better comparisons with contemporary studies. Authors need to provide specific attention towards crediting the owner of ideas and techniques by using citations. I believe the manuscript will be suitable for publication in the journal "Diagnostics" only after solving the required significant revisions.

Major points:

1.      The abstract needs to be rewritten. There have been many vague statements. Processing images with AI to detect breast cancer has been studied extensively, so making comments about existing research and limitations needs to be very specific. Also, what the authors have proposed needs to be justified using a few sentences. "hybrid optimization" or "enhanced deep learning features" are not specific.

2.      Upto line 101, the introduction is very generic. Authors are suggested to bring the necessity of new deep learning techniques for this problem and introduce much earlier what prompted them to use newer techniques. The discussion authors had between 103-110 should be extended. They used an introduction and the very shortend form of lines 1-101.

3.      The authors proposed a few challenges in lines 111-117. Did the authors solve these challenges in this manuscript? Also, "feature engineering" can not be tagged as a challenge. There has been much research on improving feature engineering in this topic. Authors need to prove that feature engineering is a challenge before using it.

4.      Before listing significant contributions (line 118), authors need to state why they used those techniques; there has to be a rationale behind their approach.  

5.      Why authors did not mention other high-performing studies and limit the discussion to CNN-based techniques?

6.      The authors concluded that the listed studies in section 2 missed "optimization of extracted features" since this is a common step in implementing deep learning.

7.      Before proposing the framework, authors must explain logically why they chose EfficientNet-b0.

8.      Does the author think the dataset (they have used) size is sufficient considering their mentioned challenge with dataset size earlier?

9.      Use citation of the dataset for figure 2,3

10.   The authors did not provide any citation for equations 1,2,3,4,5, and so on. So, does the author claim the novelty of the technique and the mathematical model?

11.   Data augmentation is not a novel or new technique; it has been used extensively in research. Authors must clearly explain why and how they used augmentation and why they think their implementation differs from other implementations already done in the literature survey.

12.   If the author does not own or create any equation or figure, they need to cite correctly. I do not see any citation for the equation or figures.

13.   Authors need to explain the reason or rationale behind choosing "equilibrium-Jaya controlled Regula Falsi" rather than explaining the technique.

14.   Table 10 needs more studies emphasizing those that performed better and used advanced deep-learning techniques. The discussion needs to be revised based on those changes.

15.   There has to be a section with limitations of the experiment presented in this manuscript. 

Author Response

Dear Editor, Response sheet has been attached. thank you

Author Response

Dear Editor, Response sheet is attached.  thank you

Round 2

Reviewer 1 Report

Missing answer to question 11. The authors have made extensive revisions and answered all the other queries satisfactorily. 

Reviewer 2 Report

Authors have made all the changes as suggested.

Round 3

Reviewer 2 Report

It can be accepted.